# Seismic Isolation Materials for Bored Rock Tunnels: A Parametric Analysis

**Ahmed Elgamal** [1,*] and **Nissreen Elfaris** [2]

1    Faculty of Engineering, Damietta University, Damietta 34517, Egypt
2    National Authority for Tunnels, Cairo 11511, Egypt
*    Correspondence: aelgamal@du.edu.eg

**Abstract:** Most recent tunnel designs rely on more thorough analyses of the intricate rock interactions. The three principal techniques for excavating rock tunneling are drill-and-blast for complete or partial cross-sections, TBM only for circular cross-sections with full faces, and road header for small portions. Tunnel-boring machines (TBM) are being utilized to excavate an increasing number of tunnels. Newer studies have demonstrated that subterranean structures such as tunnels produce a variety of consequences during and after ground shaking, challenging the long-held belief that they are among the most earthquake-resistant structures. Consequently, engineering assessment has become crucial for these unique structures from both the geotechnical and structural engineering standpoints. The designer should evaluate the underground structure's safety to ensure it can sustain various applied loads, considering both seismic loads and temporary and permanent static loads. This paper investigates how adding elastic, soft material between a circular tunnel and the surrounding rock affects seismic response. To conduct the study, Midas/GTS-NX was used to model the TBM tunnel and the nearby rock using the finite element (F.E.) method to simulate the soil–tunnel interactions. A time–history analysis of the El Centro (1940) earthquake was used to calculated the stresses accumulated in the tunnels during seismic episodes. Peak ground accelerations of 0.10–0.30 g, relative to the tunnel axis, were used for excitation. The analysis utilized a time step of 0.02 s, and the duration of the seismic event was set at 10 s. Numerical models were developed to represent tunnels passing through rock, with the traditional grout pea gravel vs. isolation layer. A parametric study determined how isolation material characteristics like shear modulus, Poisson's ratio, and unit weight affect tunnel-induced stresses. In the meantime, this paper details the effects of various seismic isolation materials, such as geofoam, foam concrete, and silicon-based isolation material, to improve protection against seismic shaking. The analysis's findings are discussed, and how seismic isolation affects these important structures' performance and safety requirements is explained.

**Keywords:** seismic performance; rock; soil–structure interaction; pea gravel; isolation materials

## 1. Introduction

Tunnel construction is an essential part of urban infrastructure. Recent seismic data show that tunnels are vulnerable to long-term seismic damage [1–9]. Tunnel damage is determined by the surrounding soil, the tunnel's submerged depth, the earthquake magnitude, and the groundwater table (G.W.T.) conditions. Tunnels often behave differently during seismic activity depending on the kind of soil in which they are constructed. While tunnels constructed in solid rock are far less likely to sustain substantial damage, tunnels constructed in soft soil are significantly more likely to sustain damage.

Shallower tunnels are more likely to undergo deformation when subjected to seismic waves than deeper tunnels [10–20]. Scholars have carried out numerical and experimental investigations into the seismic activity of various components of underground structures. Utilizing a Cambridge University centrifuge facility, the influence of circular cross-section tunnels on the nearby ground's acceleration response was researched by Lanzano et al. [21].

In their trials, they looked at the impacts of tunnel depth and soil density. Tunnel seismic response was defined by Owen and Scholl [22] via tunnel deformation modes, including ovaling, racking, curvature, and axial deformation. Wang [23,24] and Penzien [25] introduced closed-form analytical techniques to determine different straining actions produced in the lining layer as well as to evaluate the effects of ovalization and racking. Through a series of centrifuge model experiments, Cilingir and Madabhushi [26–28] investigated tunnels' seismic behavior in dry sand in order to ascertain the effects of entry soil depth and motion. Several analytical [1,27,28], numerical [29–33], and experimental [34–37] studies have shed light on tunnel seismic behavior. According to this study, tunnel damage resulting from earthquakes is expensive and challenging to repair; therefore, adequate disaster-avoidance strategies must be incorporated into the seismic analyses of tunnels.

Foam concrete has the capability to conduct seismic isolation for tunnels due to its critical energy-absorption capacity, as investigated by Ma et al. [38]. This research aimed to investigate the thermal insulation effects of the foamed concrete layer within the rock tunnel and its mechanical properties. The effects of density, confining stress, and strain rate on the mechanical characteristics of foamed concrete were investigated through experiments. Triaxial and uniaxial compression tests were conducted. The effects of concrete density and normal stress on the non-linear behavior of the foamed concrete layer–lining interface were studied using a direct shear test. The results of the tests indicated that the concrete's density substantially influences the foamed concrete's mechanical properties. Furthermore, the volumetric compressibility and dependency on the strain rate of foamed concrete are noteworthy characteristics.

The isolation layer used against seismic effects is one method that alleviates soil restrictions and decreases structural distortions resulting from seismic loadings. However, this unique technology is not well acknowledged in subterranean structures in soft ground. When investigating the effects of foamed concrete on the lining of a tunnel in rocky ground as a seismic isolation layer, Li and Chen [39] applied an F.E. technique to determine the results of their investigation. In addition, investigations were conducted into the layer's density and thickness and the interface's shear stiffness and residual friction coefficient. It was found that a foamed concrete layer characterized by a reduced shear stiffness or residual friction coefficient, coupled with a high thickness and low density, effectively enhanced seismic isolation.

Many scholars [40–47] have examined the seismic isolation effect of a layer of foamed concrete on rocky soil using a 3D F.E. model. Their findings have been published in the journal *Earthquake Engineering*. The tunnel lining's behavior relative to the rock deformation was found to be lessened due to the shear deformation of the layer of foamed concrete, which took up the shear force that moved from the surrounding rock to the tunnel's body. EPS geofoam is a lightweight yet durable material made of expanded polystyrene via a process known as polymerization. Its closed-cell structure makes it resistant to water and decomposition, with virtually no need for maintenance.

Utilizing geofoam as a layer of isolation between the tunnel liner and the rock [48–51] constructed an experimental structure. Dynamic ground pressure was lowered by 70% to 90% in the seismic isolation layer compared to a structure without one. Tunnels constructed in compacted rock endured less damage than those constructed in malleable soil [13,52–55]. In contrast to surface structures, tunnel dynamics are predominantly governed by deformations of the adjacent soil as opposed to inherent vibration properties. Consequently, a comprehensive investigation is required to determine the impact of a soft soil isolation layer on a tunnel's seismic response.

Many researchers [56–59] have tried to evaluate the possibility of using asphalt and cement (A-C) materials as a seismic isolation layer while backfilling grouting shield tunnels. In this research, experimental procedures that were both standardized and comparative were carried out in order to evaluate the grouting characteristics of A-C materials. The trials' conclusions indicated that the fluidity and consistency of A-C materials, i.e., the changes in cement content brought about by increasing cement content, are significant. As

the concentration of cement increased, so did the ultimate compressive strength; therefore, it was advised to use a mass ratio of 50% cement to 50% asphalt for the seismic isolation layer. The study focused on a planned shield tunnel to cross a hard–soft stratum. The tunnel lining's seismic response was evaluated using numerical simulations at different points along the reinforced A-C seismic isolation layer. According to the computation findings, the tunnel lining was hard–soft. In close proximity to the stratum junction, the A-C seismic isolation layer was found to considerably reduce the maximum main stress of the lining. Despite this, the strata intersection remained vulnerable to seismic damage. For the shield tunnel under consideration, the A-C isolation layer's realistic reinforced length was three times the tunnel diameter measured from the intersection of the strata on all sides.

## 2. Rock Tunneling

The three principal excavation techniques used in rock tunneling are drill-and-blast for any cross-section, TBM for only circular cross-sections, and road header for partial face progress, any cross-section, or full-face for small sections. The cost of TBM is more than that of drill-and-blast and road header by an approximate range between 20% and 35% per kilometer, inclusive of all design, construction, and material costs associated with tunneling according to the geology, the tunnel diameter, and regions [60]. These approaches are often employed separately but can be used in tandem. The deformability or stiffness of the support materials, the degree of bonding between the support and the rock mass, and the installation time all influence the support materials' structural behavior. Rock reinforcement uses rock bolts, spiling or foreboding, pre-injection, steel ring beams with or without lagging, inverted segments, shotcrete, precast concrete segmental liner, and other ground-support elements used in rock tunneling. Significant breakthroughs in TBM technology and related sectors for rock tunneling have occurred in recent years. As a result, TBM technology can start and finish projects on time, even under challenging conditions. The shielded hard rock TBM's fundamental idea is taken from its use in soil. The excavation rate by D and B is 3–9 m/day, but by TBM, it is 15–50 m/day [61].

The primary distinction lies in the cutting instruments employed and fixed to the cutter head, even with the secondary requirement for facial support. The cutter head is fitted with disc cutters that roll in concentric tracks and are pressed against the face of the hard rock. They release chips from the tunnel face as the cutter head rotates, achieving a steady advance rate throughout the "boring" process (Figure 1).

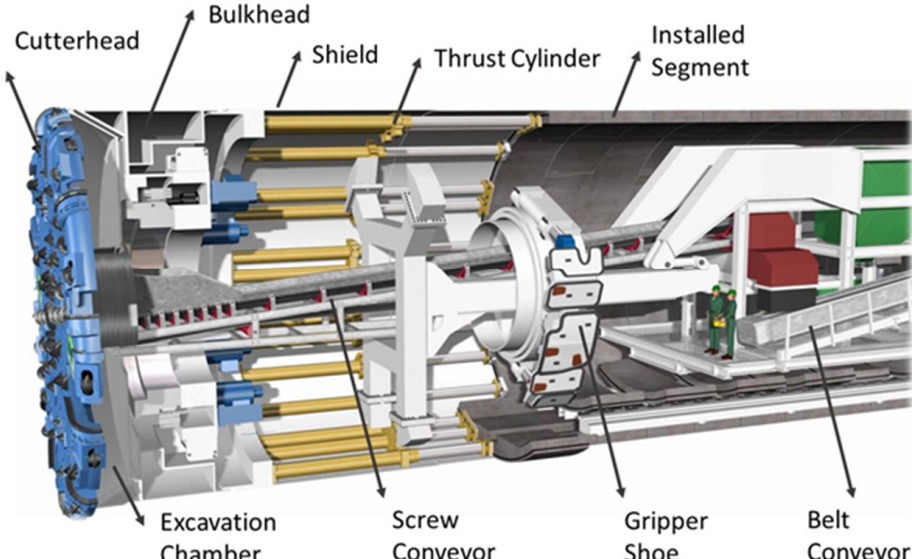

**Figure 1.** Schematic view of a double-shield TBM.

Grouted pea gravel is used in shielded tunnel-boring machine (TBM) tunnel construction to fill spaces between segment rings and surrounding rock. As a connecting layer, this filler layer significantly impairs the segment ring's self-sustenance. The backfilling layer facilitates the load transfer between the lining and the rock mass; the model analyzes and represents the subsequent interaction. Dry mortar is utilized to fill the space at the invert of a tunnel dug in "hard rock" to support the ring. Following the injection of cement grout, pea gravel is applied along the remaining surface. In the context of spalling, it is of the utmost importance to adequately fill this void, which may become even more substantial due to the notches caused by stress. It is postulated that the mortar-injected pea gravel has an elastic modulus of 1.00 GPa. Through the shield tail, a grout annulus is injected using a mortar that hardens incredibly quickly, while the shield advances concurrently (Figure 2).

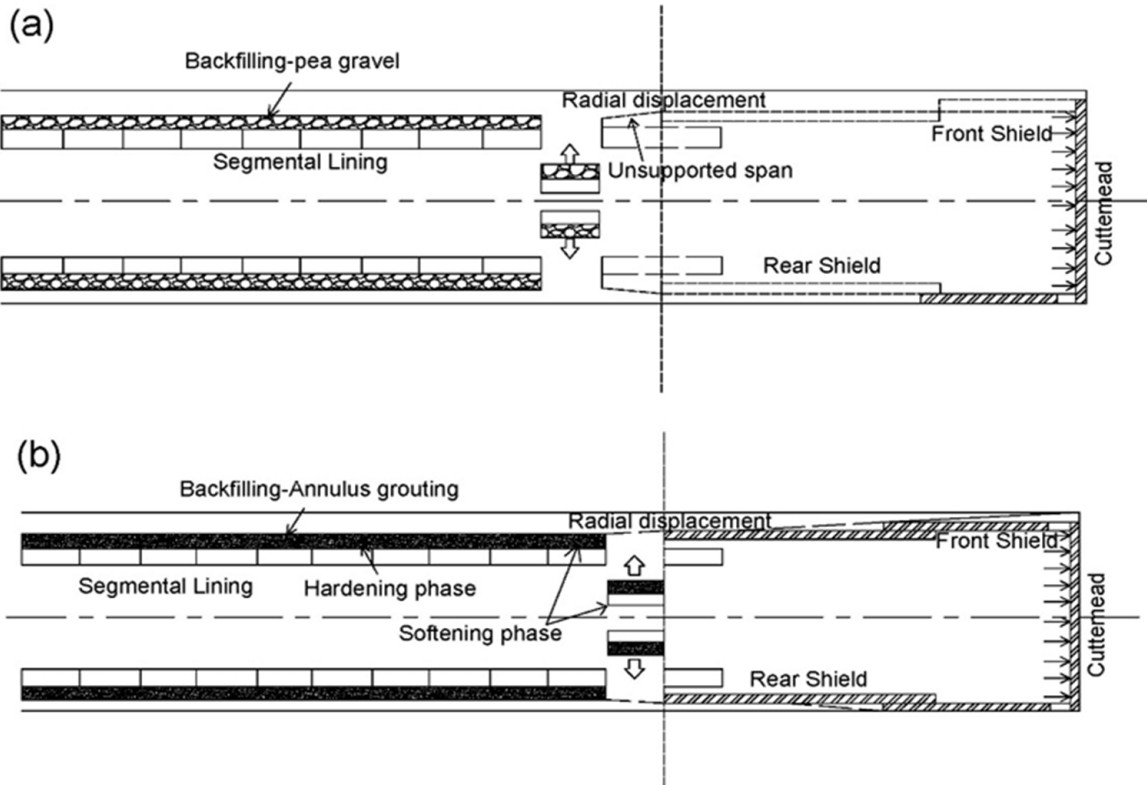

**Figure 2.** Backfilling of the segmental lining: (**a**) with pea gravel and (**b**) with annulus grouting via the shield tail.

### 2.1. Seismic Response of Rock

Rock is affected by seismic waves as they pass through it. Primary waves oscillate parallel to the direction of motion of the wave. They are sometimes referred to as compression waves or longitudinal waves. Primary waves frequently arrive first because they go through the crust faster, which is how they received their name. They do not cause much damage because they arrive from below and make structures move vertically.

The secondary waves, also known as S-waves, emerge after some time, usually a few seconds, although this can change. These waves are transverse, meaning their displacement is perpendicular to the wave's direction of motion, which causes the ground to tremble. Since structures are significantly weaker during this oscillation, these waves are primarily responsible for most damage and the fatalities and injuries brought on by structure collapse. Figure 3 shows the direction of the body waves produced by earthquakes.

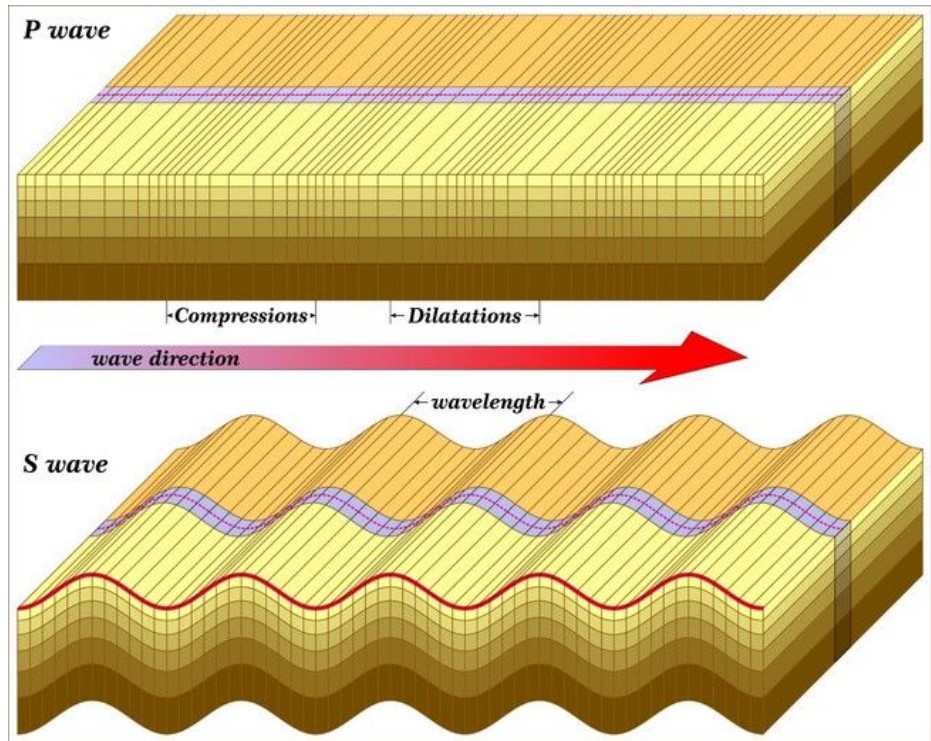

**Figure 3.** Direction of body waves (P-waves and S-waves generated by earthquakes).

Research on the relationship between shaking and distance has revealed that waves lose energy as they travel. Depending on the kind of rock they are passing through, they can lose energy at different rates. For example, extremely solid granites are better at transmitting energy than the broken-up rocks found in fault zones. It almost seems the waves are losing more energy—at exceptionally high frequency—as they pass through the broken rocks because they move further.

Hard rocks allow seismic waves to propagate faster than weaker sediments and rocks. As the energy builds up, the waves flow from deeper, harder rocks to shallower, softer rocks, slowing down and increasing amplitude. The wave size increases with the softer rock or soil beneath a site. Ground motion is amplified on softer soils. Elastic rebound is when the rock returns to its original shape. The asperities stop rocks from slipping on each side of an active fault. The rock is distorted by stress until the asperities fracture, releasing the tension and allowing it to return to its natural form. Three stages of deformation occur in a rock when it is subjected to increasing stress, as shown in Figures 4 and 5.

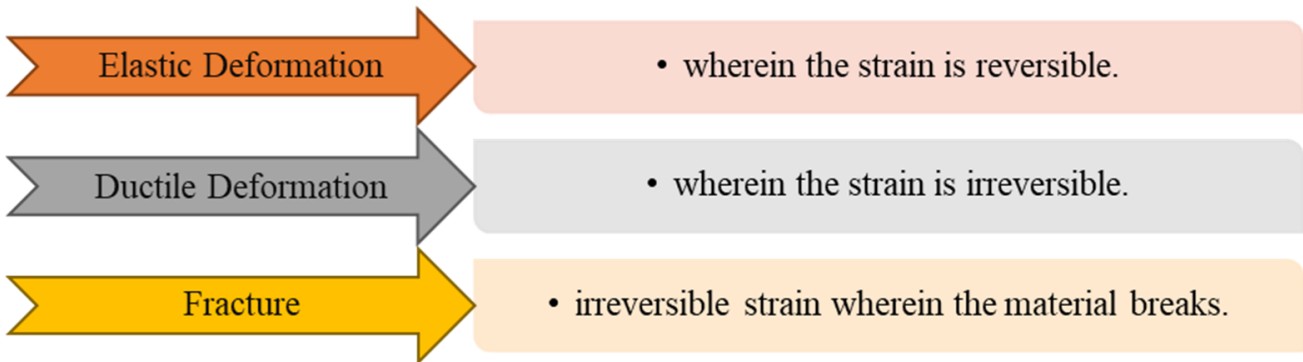

**Figure 4.** Stages of deformation in rock.

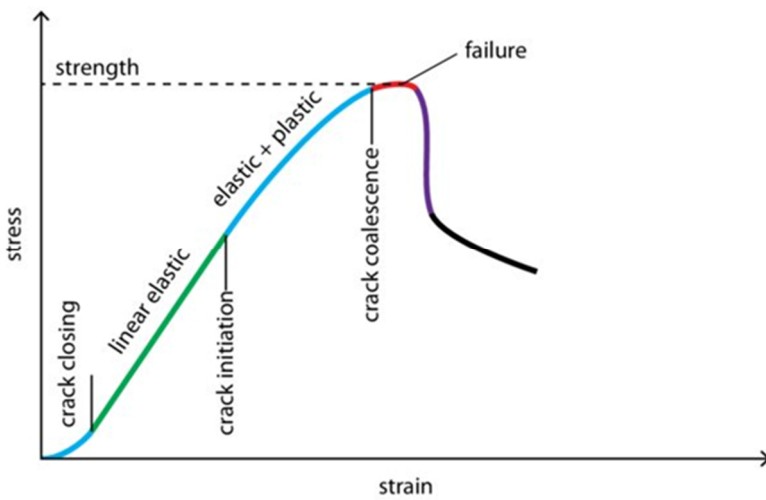

**Figure 5.** The typical stress–strain curve for rock.

### 2.2. Seismic Isolation of Rock Tunneling

Numerous strategies exist to lower the stresses that ground deformation causes in tunnels. One method is adding an isolating substance between the tunnel and the surrounding rock. Because of their high energy-absorption capacity, certain materials can be used to isolate rock tunnels from seismic activity. The surrounding earth's stiffness must be taken seriously compared to the tunnel lining to decrease stress concentrations in the tunnel. Added tunnel isolation materials and adjusted liner stiffness make it stronger and more ductile.

A flexible seismic isolation grout protects a tunnel from surrounding ground deformations. It reduces internal stresses by an approximate range of 40% to 60% and more when applied to the outer border of the tunnel [61]. The same construction methods used to backfill grout into the tail vacuum also apply with shield-driving tunnels or injecting material from within the tunnel, as seen in Figure 6. Materials for seismic isolation suitable for underground construction must serve as backfill materials.

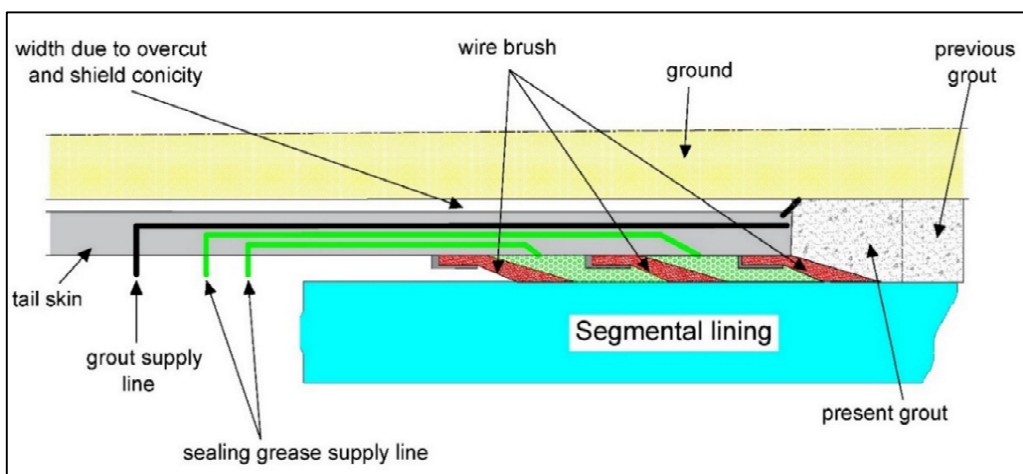

**Figure 6.** Grouting through tail skin [62].

As shown in the following expressions, the stiffness of the isolation layer, which is proportional to the shear modulus (*G*) and Poisson's ratio (*v*) in the longitudinal and transverse axes of the tunnel, determines the seismic isolation mechanism. This is the mechanism that determines seismic isolation.

$$K_x = \frac{2\pi G}{\ln R / r} \tag{1}$$

$$K_y = \frac{8\pi G(3-4v)(1-v)}{(3-4v)^2 ln\left(\frac{R}{r}\right) - \left[\left(\frac{R}{r}\right)^2 - 1/\left(\frac{R}{r}\right)^2 + 1\right]} \tag{2}$$

where $K_x$—stiffness coefficient of the isolation layer along the tunnel's longitudinal axis; $K_y$—stiffness coefficient of the isolation layer in the tunnel's transverse direction; $R$—isolation layer outer diameter; $r$—isolation layer inner diameter; $G$—shear modulus of the isolation layer; $v$—Poisson's ratio of the isolation layer.

## 3. Numerical Modeling

It is critical to realize that not all tunnels can be evaluated similarly. Furthermore, the precision of the analytical techniques currently in use is significantly greater than the accuracy and reliability of site-inspection data [63,64]. As a result, designers must undertake many diverse tests to understand the ground–support interaction model's sensitivity to input parameters. During this step, designers should employ various design techniques to identify the design parameters, upper and lower boundaries, and the design's sensitivity to diverse aspects.

The accessible techniques are empirical, "closed-form" methods of analysis and numerical methods. Numerous intricate aspects of tunnels can be explicitly simulated by numerical techniques such as those that use the discrete element (D.E.), boundary element (B.E.), finite difference (F.D.), and F.E. methods.

The main objective of this study is to demonstrate that the commercial 3D F.E. method tool Midas/GTS-NX ver. 1.1 may be utilized to perform 3D numerical modeling to predict the behavior of soil–tunnel interactions. The current study attempted to optimize the design by lowering the strains imposed on tunnels by seismic waves using isolation material to absorb their dissipation energy.

*Model Establishment*

The present investigation examines a rock-encased, deep-bored tunnel of 9.30 m in diameter and 450 mm in lining thickness, as seen in Figure 7. The tunnel is 25 m below the surface of the ground (H) and has a section width of 1.5 m. As illustrated in Figure 8, The dimensions of the overall model are 150 m × 90 m × 60 m in the x, y, and z axes. Solid components were selected for the isolation coating: pea gravel grout, tunnel, and rock. A thickness of 200 mm of the grout or isolation coating was utilized to occupy the shield machine's tail void. The ground was postulated to comprise a single rock stratum, as illustrated in Figure 8. The behavior of the soil is characterized by the non-associated Mohr–Coulomb (MC) standards. A list of the soil's material properties may be found in Table 1.

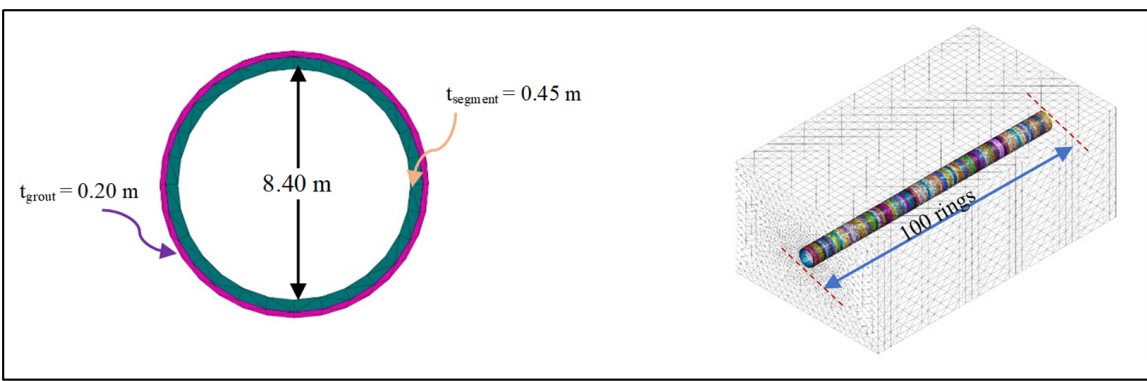

**Figure 7.** Tunnel and grout numerical model configuration.

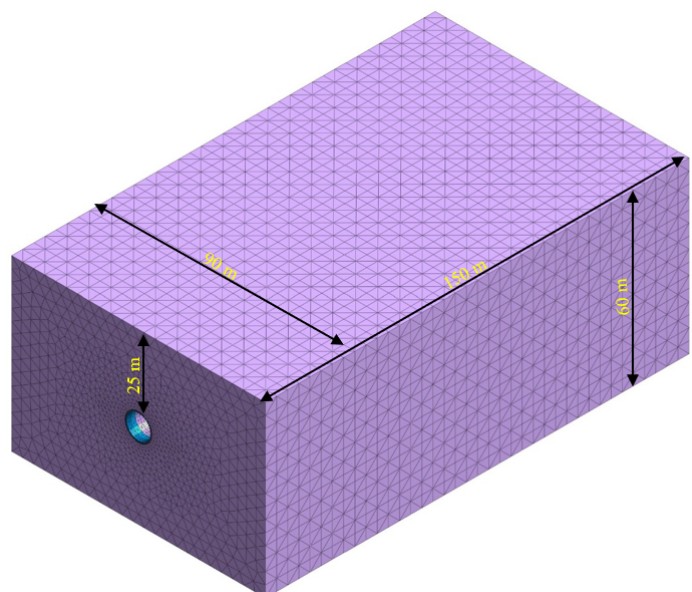

(**a**) One rock layer.

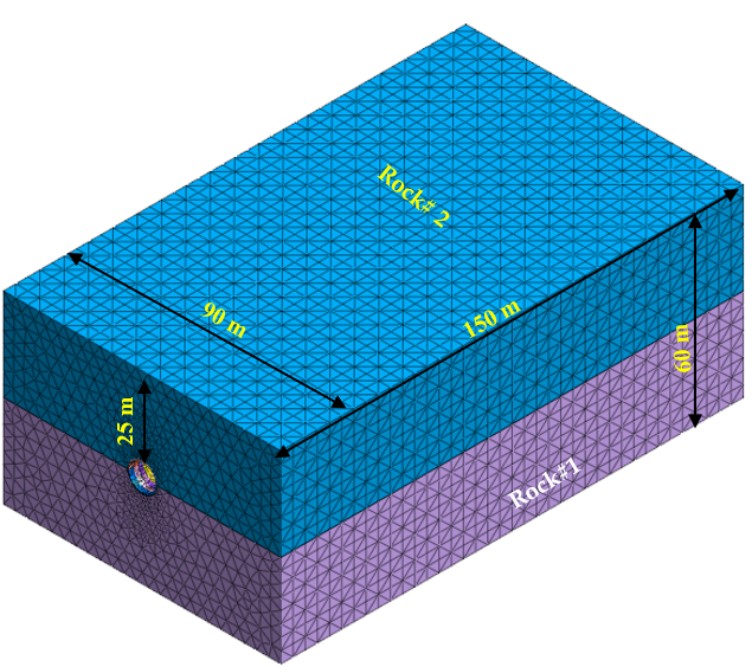

(**b**) Two rock layers.

**Figure 8.** F.E. numerical model boundaries.

**Table 1.** Numerical model properties of rock.

| Material | Modulus of Elasticity (E) (MPa) | Poisson's Ratio ($\nu$) | Unit Weight ($\gamma$) (kN/m$^3$) | Cohesion ($c_u$) (kN/m$^2$) | Friction Angle ($\varphi$) (°) |
|---|---|---|---|---|---|
| Rock #1 (medium strong) | 6.0 | 0.30 | 23.00 | 700 | 39° |
| Rock #2 (medium weak) | 1.0 | 0.30 | 23.00 | 200 | 30° |

Table 2 presents the elastic characteristics of the tunnel as well as the typical grout materials that were utilized in the model.

**Table 2.** Characteristics of the materials used for grout and concrete lining.

| Material | Modulus of Elasticity (E) (MPa) | Poisson's Ratio (ν) | Unit Weight (γ) (kN/m³) |
|---|---|---|---|
| Concrete for segment | 30.0 | 0.20 | 25.00 |
| Grout for the tunnel (pea gravel) | 1.0 | 0.30 | 23.00 |

Midas/GTSNX ver. 1.1. [65] is a fully integrated F.E. analysis software for geotechnical engineering applications. This F.E.-based program was developed to evaluate soil–structure interaction. Midas/GTS-NX helps engineers perform step-by-step analyses of excavation, banking, structure placement, loading, and other factors directly affecting design and construction. The program supports various conditions (soil characteristics, water level, etc.) and analytical methodologies to simulate natural phenomena. Settings for all field conditions can be simulated using non-linear analysis methods such as linear and non-linear static analysis, linear and non-linear dynamic analysis, seepage and consolidation analysis, slope safety analysis, and various coupled analyses.

Midas/GTS-NX program can automatically constrain the model. The nodal degrees of freedom (D.O.Fs) are restricted in the x-direction along the model's vertical sides. The front and back sections of the model are confined in the y-direction. This means that the degrees of freedom (D.O.Fs) along the z-axis are restricted in the y and x axes for the lowest nodes. There is no restriction on the degrees of freedom along the ground's surface. An application of linear dynamic time history analysis uses the direct integrating method. These are the El Centro 1940 data utilized as dynamic excitation sources that were not parallel to the tunnel axis and were obtained from the Midas/GTS-NX library (Figure 9). The acceleration at ground maximum is 0.30 g. In this investigation, the excitation period is 10 s, and each time step is 0.02 s or 500 steps.

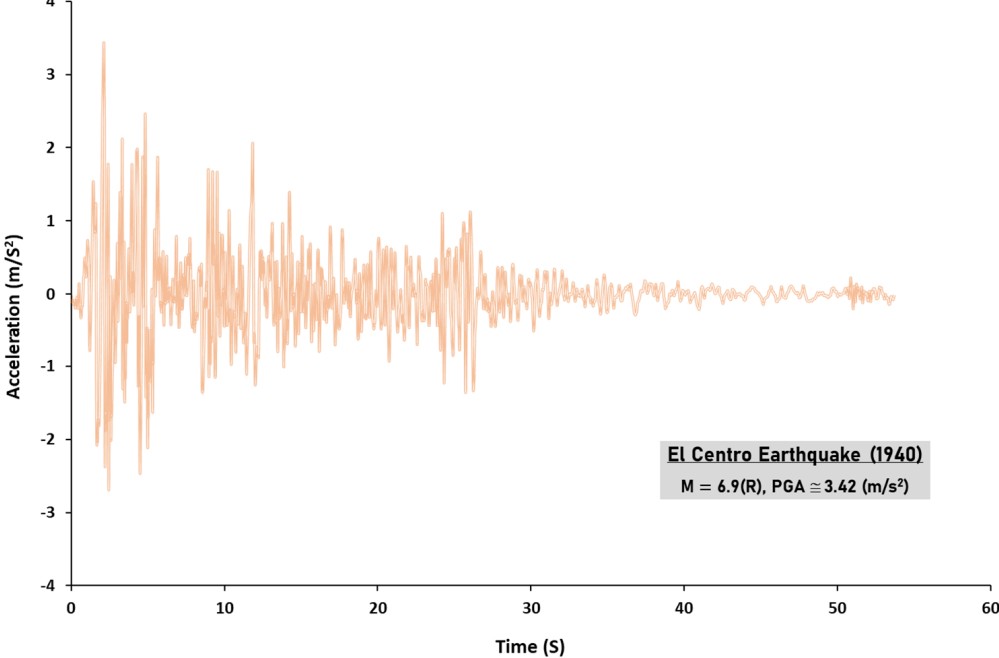

**Figure 9.** Acceleration time history.

The El-Centro-affected areas were in the United States and Mexico. The El Centro earthquake is significant in seismic design because it was one of the first to record comprehensive, robust motion data, which is one of the main reasons it is regarded as a reference earthquake. Linear dynamic time–history analysis using a direct integrating method was applied. The direct integration method analyzed all time stages, and the number of time stages is proportional to the analysis time.

Seventeen F.E. numerical models were considered to check the tunnel's stability under seismic loading, as shown in Figure 10. Experts have studied the effect of changes in isolation materials properties, changing the unit weight of isolation to rock, the Poisson ratio of isolation, and the shear modulus of isolation to rock ratios.

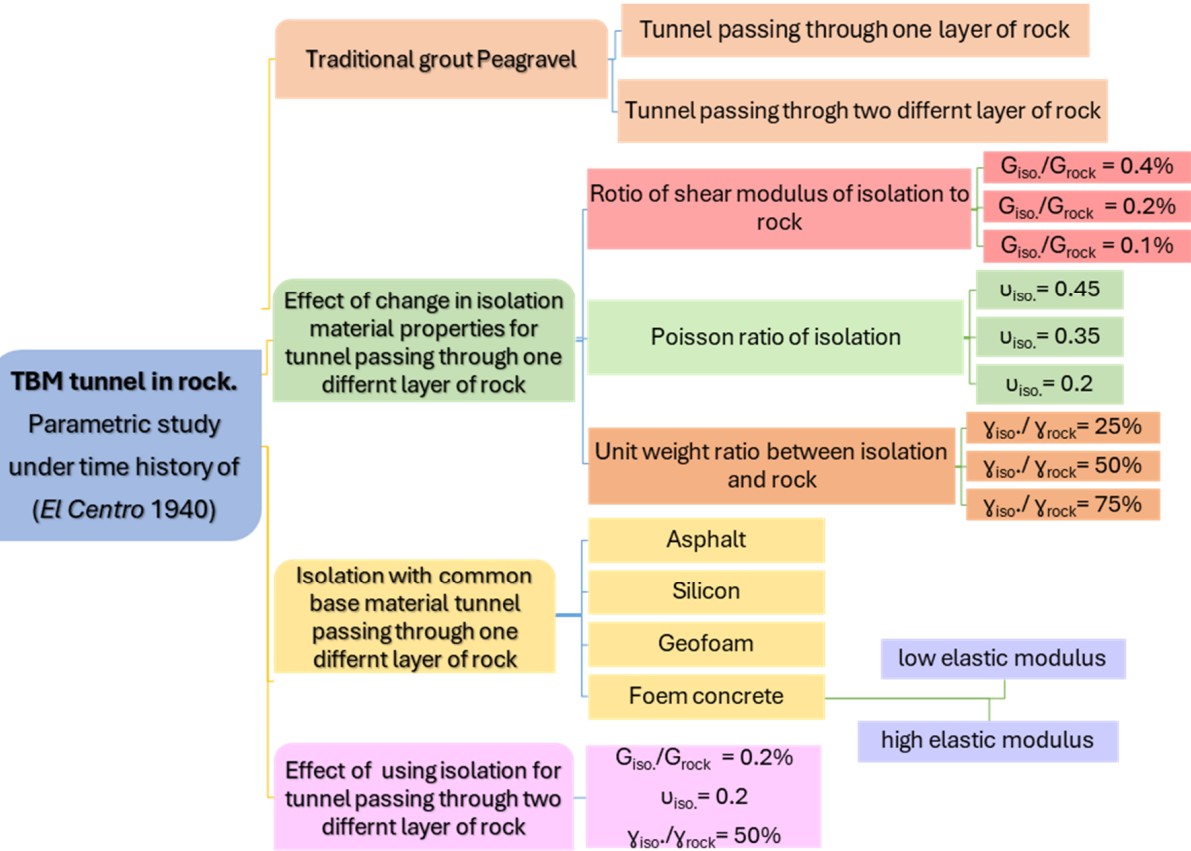

**Figure 10.** Flowchart for the creation of numerical modeling.

Separated into two parts is the parametric analysis that was conducted for the isolation material that was utilized in the model: The first section examines the effects of changes in the isolation material's mechanical properties, such as shear modulus, poison's ratio, and unit weight assumed concerning the mechanical properties of the rock, on the stresses conferred upon the tunnel, as shown in Table 3. The second section examines common-based materials such as silicon, asphalt, geofoam, and foamed concrete with low and high elastic modulus values on the tunnel's stresses, as shown in Table 4.

**Table 3.** Isolation material properties.

| Material | Modulus of Elasticity (E) (kN/m²) | Poisson's Ratio (ν) | Unit Weight (γ) (kN/m³) |
|---|---|---|---|
| $G_{iso}/G_{rock} = 0.4\%$ | 25,000 | 0.35 | 11.50 |
| $G_{iso}/G_{rock} = 0.2\%$ | 12,500 | 0.35 | 11.50 |
| $G_{iso}/G_{rock} = 0.1\%$ | 6250 | 0.35 | 11.50 |

**Table 3.** *Cont.*

| Material | Modulus of Elasticity (E) (kN/m²) | Poisson's Ratio (ν) | Unit Weight (γ) (kN/m³) |
|---|---|---|---|
| $\nu_{iso} = 0.45$ | 12,500 | 0.45 | 11.50 |
| $\nu_{iso} = 0.35$ | 12,500 | 0.35 | 11.50 |
| $\nu_{iso} = 0.20$ | 12,500 | 0.20 | 11.50 |
| $\gamma_{iso}/\gamma_{rock} = 25\%$ | 12,500 | 0.35 | 5.75 |
| $\gamma_{iso}/\gamma_{rock} = 50\%$ | 12,500 | 0.35 | 11.50 |
| $\gamma_{iso}/\gamma_{rock} = 75\%$ | 12,500 | 0.35 | 17.50 |

**Table 4.** Property characteristics of materials commonly used for isolation.

| Material | Modulus of Elasticity (E) (kN/m²) | Poisson's Ratio (ν) | Unit Weight (γ) (kN/m³) |
|---|---|---|---|
| Silicon | 500 | 0.48 | 12.00 |
| Geofoam | 4800 | 0.10 | 0.16 |
| Foamed concrete (low E) | 44,720 | 0.35 | 3.00 |
| Foamed concrete (high E) | 760,000 | 0.22 | 7.30 |
| Asphalt | 101,400 | 0.30 | 9.60 |

## 4. Results

In this study, the main focus is on the stresses generated in the transverse direction. The tunnels are constructions designed to withstand movements in a transversal approach. In order to arrive at correct findings for actual design work, the earthquake time history utilized in this study is believed to create the most significant reaction since it produces the highest response in terms of absolute stresses. Table 5 presents a comparison of the stresses on the tunnel by using typical grout and by isolating the tunnel.

**Table 5.** Impact of using isolation as an alternative to conventional grout.

| Grout/Isolation Descriptions | Absolute Stresses |
|---|---|
| $G_{iso}/G_{rock} = 0.40\%$ | 58–64% |
| $G_{iso}/G_{rock} = 0.20\%$ | 34–41% |
| $G_{iso}/G_{rock} = 0.10\%$ | 18–24% |
| $\nu_{iso} = 0.45$ | 67–73% |
| $\nu_{iso} = 0.35$ | 34–41% |
| $\nu_{iso} = 0.20$ | 23–31% |
| $\gamma_{iso}/\gamma_{rock} = 25\%$ | 34–41% |
| $\gamma_{iso}/\gamma_{rock} = 50\%$ | 34–41% |
| $\gamma_{iso}/\gamma_{rock} = 75\%$ | 35–41% |
| Silicon | 2–9% |
| Geofoam | 1–10% |
| Foamed concrete (low E) | 62–70% |
| Foamed concrete (high E) | 91–92% |
| Asphalt | 27–101% |

Using traditional grout and isolation with a shear modulus ratio of 0.1% to 0.4% or Poisson's ratio of 0.2% to 0.45%, a comparison of the stresses under the El Centro earthquake time history revealed that when this ratio decreases, the stresses decrease, as shown in Figures 11 and 12. These results are demonstrated by the fact that the ratio decreases. On the other hand, the change in unit weight does not affect stresses, as shown in Figure 13.

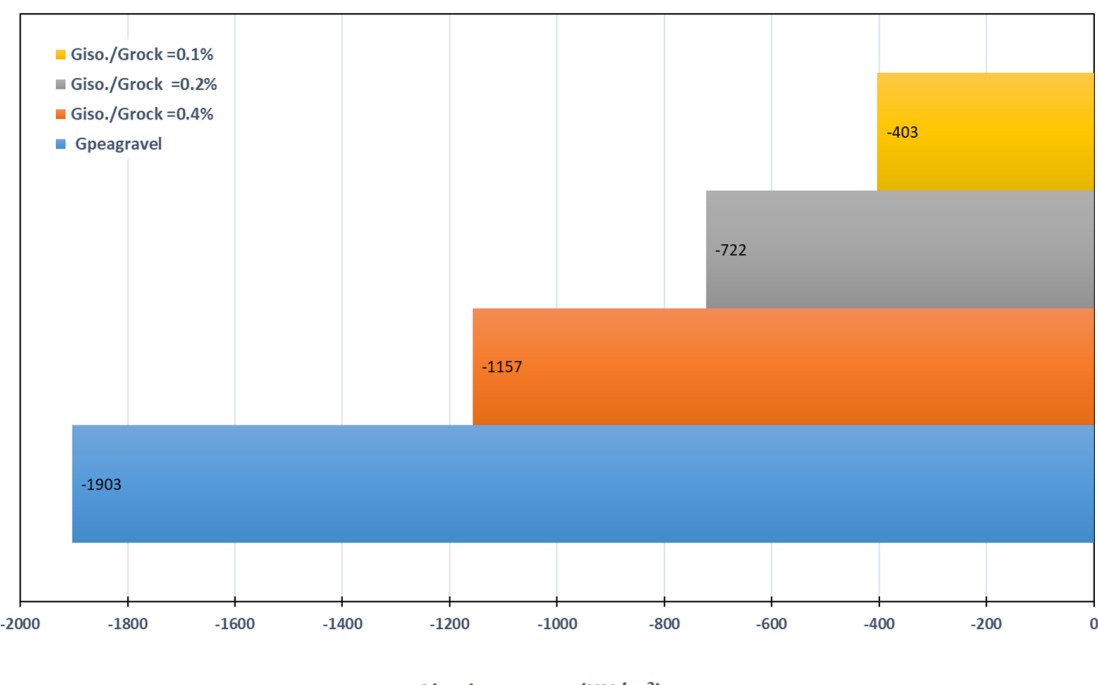

**Figure 11.** Shear modulus of isolation effect on transversal stresses in the rock tunnel.

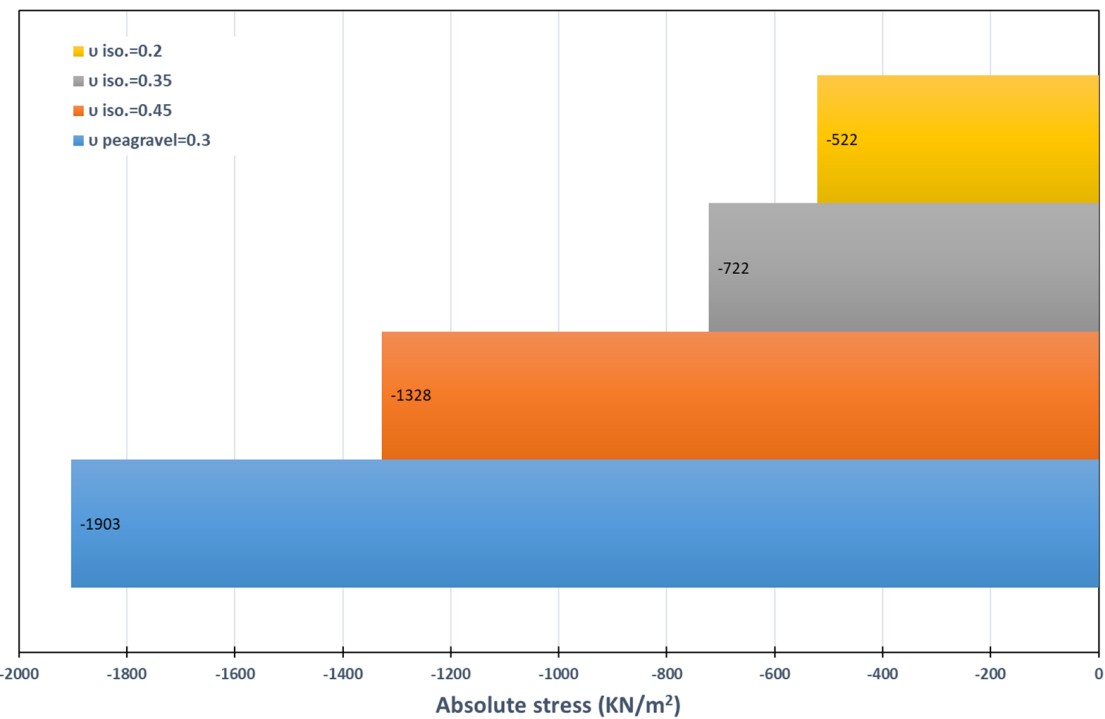

**Figure 12.** The Poisson ratio of isolation effect on transversal stresses in the rock tunnel.

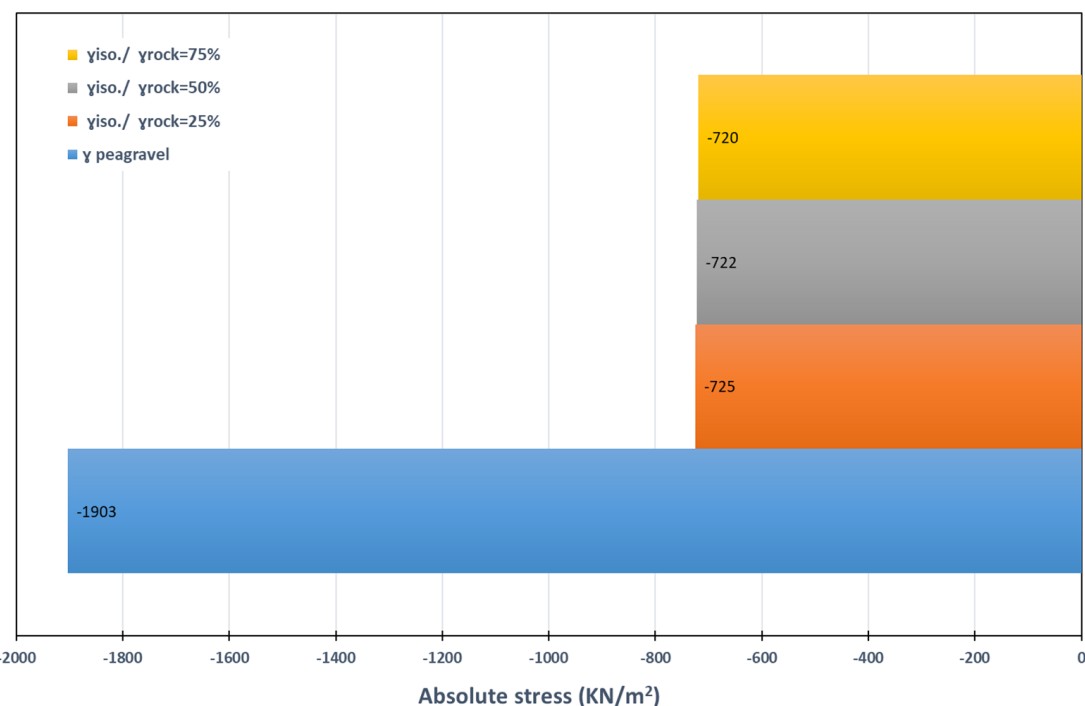

**Figure 13.** Unit weight of isolation effect on transversal stresses in the rock tunnel.

A comparison of the stresses when using traditional grout and isolation with a shear modulus ratio of 0.2% and Poisson's ratio of 0.2 for tunnel passing through two different rock layers is shown in Figures 14 and 15. The stress magnitude was thereby reduced by approximately 21% to 23% at the crown in the rock#2 layer and 34% to 46% at the inverted tunnel in the rock#1 layer.

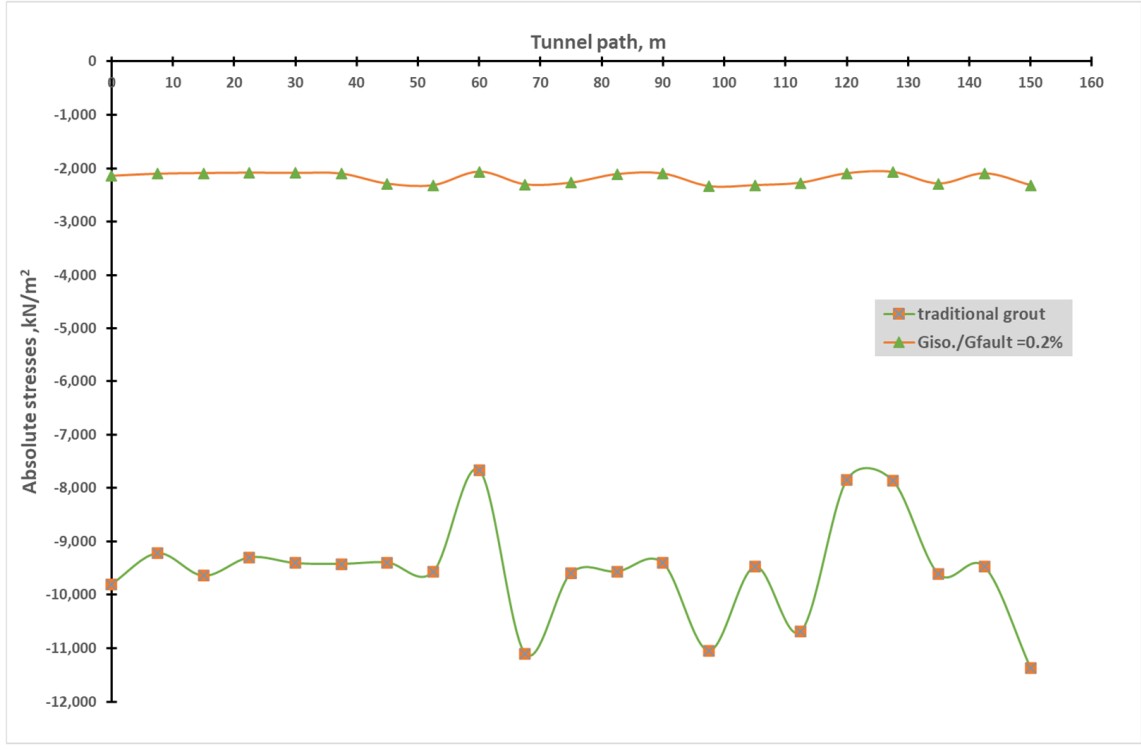

**Figure 14.** Effect of isolation on transverse stresses of the tunnel crown in two different rock layers.

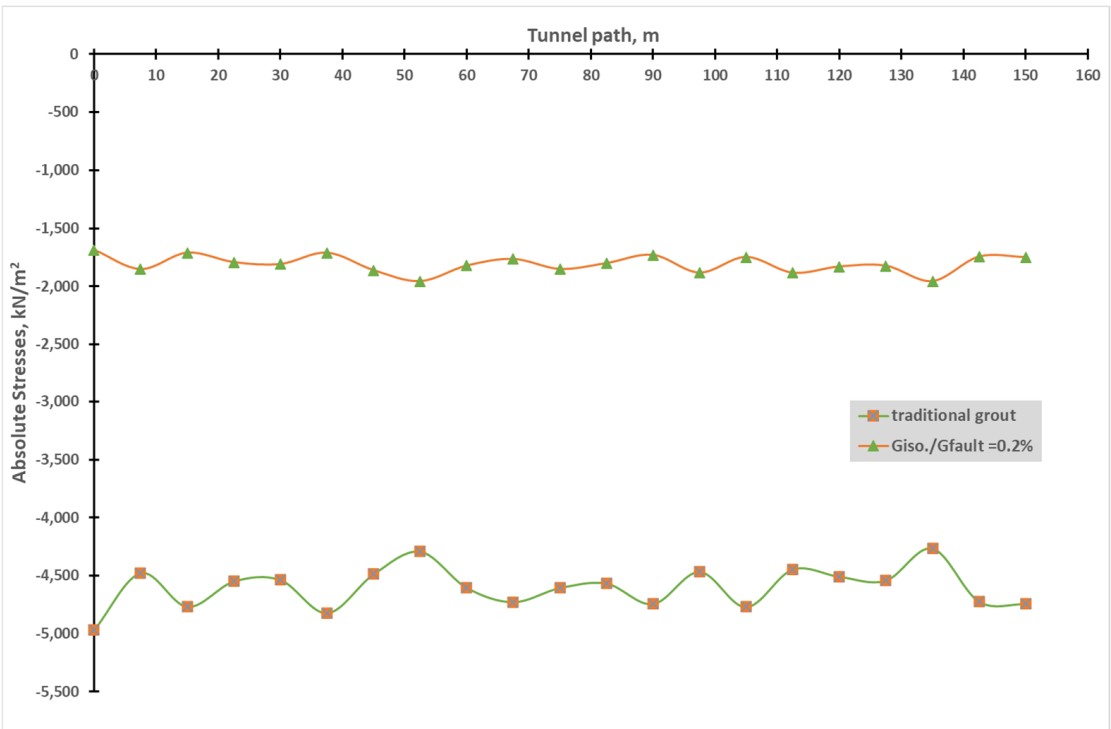

**Figure 15.** Effect of isolation on inverted tunnel transverse stresses in two different rock layers.

## 5. Discussion

When comparing the stresses under the El Centro earthquake time history between conventional grout and common isolation, the foam concrete with a high elastic modulus and asphalt base material do not significantly differ. Even if the silicon base material and geofoam reduce compression stresses, they may become tensile when combined with construction load. Conversely, low-elastic-modulus foam concrete reduces stresses to a reasonable level, as shown in Figure 16.

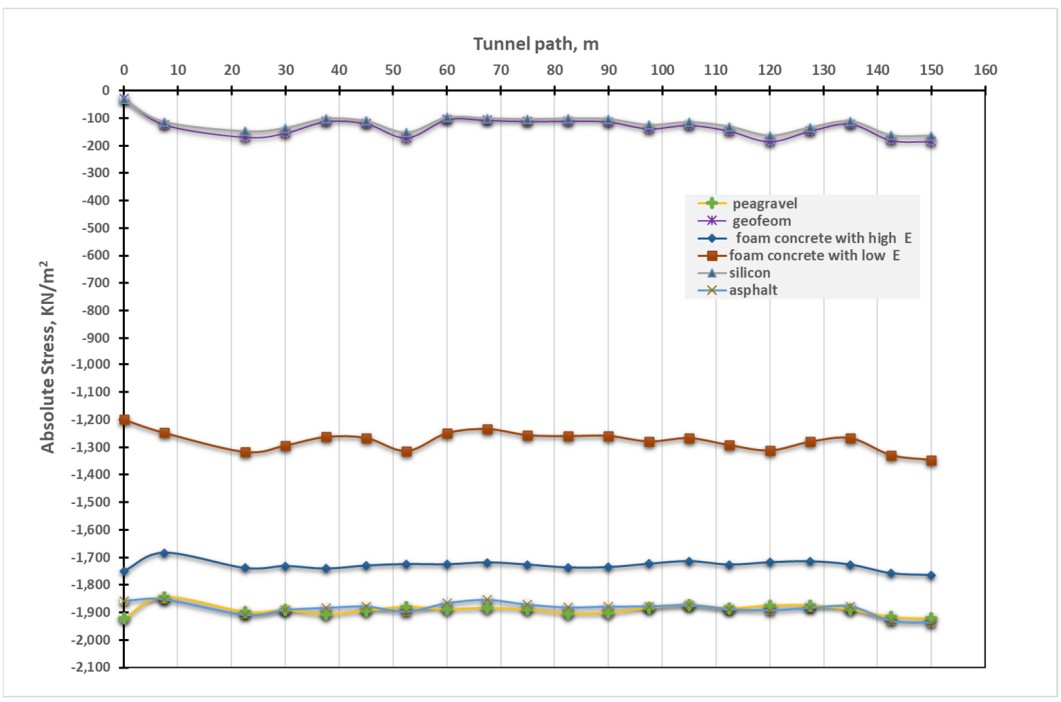

**Figure 16.** Effect of using common isolation material on a tunnel's transverse stresses in rock.

The transverse stresses of a tunnel in rock were compared regarding the effects of parametric assumption isolation and common isolation. The results showed that isolation with a Poisson's ratio of 0.2 to 0.45 or a shear modulus ratio of 0.1% to 0.4 reduces stresses to a manageable level, as shown in Figure 17.

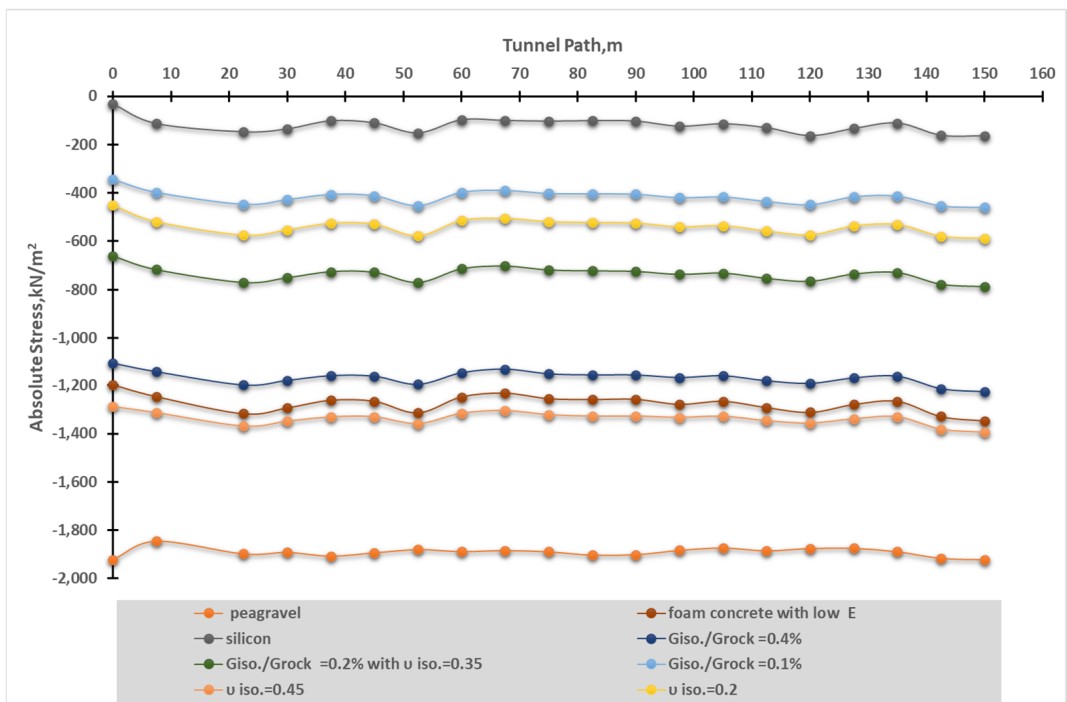

**Figure 17.** Effect of common isolation vs. user isolation on transverse stresses of a tunnel in rock.

The seismic isolation method effectively improves the seismic safety of bored tunnels. The above results show that the significance ratio between the shear modulus of isolation and the surrounding soil should be between 1/500 and 1/1000. However, the maximum values of tunnel displacement increase in the direction of seismic motion, and the dynamic behavior of the tunnel with isolation is better than with traditional grout.

When using isolation grout with a shear modulus ratio of 0.2% and Poisson's ratio of 0.2 for a tunnel passing through two different rock layers, the vertical displacement of the tunnel overall redistributes stresses (Figures 18 and 19).

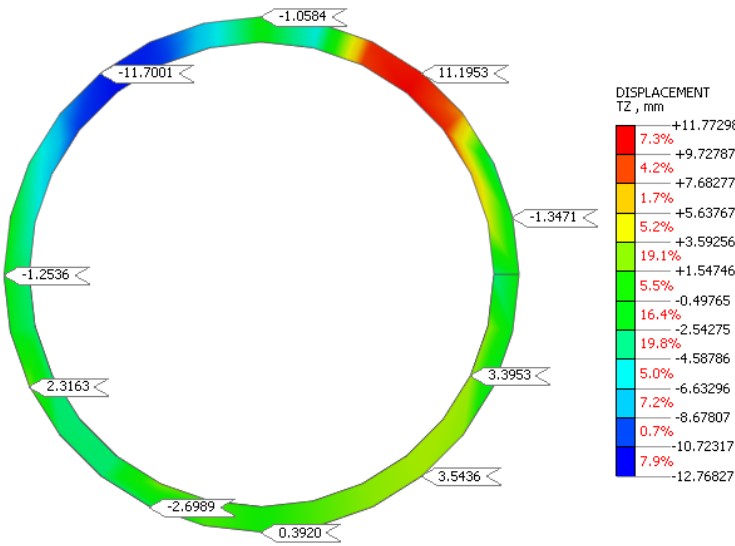

**Figure 18.** Vertical displacement of the tunnel with traditional grout.

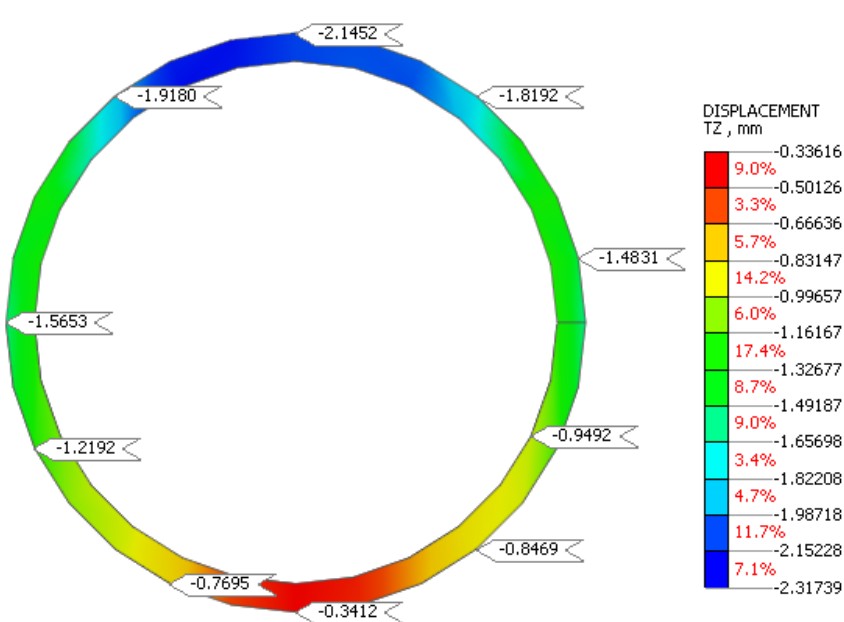

**Figure 19.** Vertical displacement of the tunnel with isolation.

## 6. Conclusions

According to seismic research conducted on a rock tunnel, isolation positively affects both kinds of stresses in the transverse directions of tunnels. There is an improvement in both the produced stresses and the overall dynamic behavior of the tunnel when isolation is employed rather than the conventional grout treatment. It is of the utmost importance to investigate the drop in displacement that occurs due to a reduction in the shear modulus of the isolation material. Construction analysis should be combined with dynamic analysis in the design process. The designers may use this approach to help them create safe and cost-effective designs. It is possible to determine whether isolation materials can withstand applied pressures without liquefying by testing them with tiny shear modules or small unit weights.

**Author Contributions:** Conceptualization, A.E. and N.E.; methodology, A.E.; software, A.E.; validation, A.E. and N.E.; formal analysis, N.E.; investigation, A.E.; resources, A.E.; data curation, A.E.; writing—original draft preparation, N.E.; writing—review and editing, A.E.; visualization, A.E.; supervision, A.E.; funding acquisition, N.E. All authors have read and agreed to the published version of the manuscript.

**Funding:** This research received no external funding.

**Data Availability Statement:** Data are contained within the article.

**Conflicts of Interest:** The authors declare no conflict of interest.

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
