# Peer review of "Seismic Isolation Materials for Bored Rock Tunnels: A Parametric Analysis"

_infrastructures, doi:10.3390/infrastructures9030044_

Round 1
Reviewer 1 Report
Comments and Suggestions for Authors
Overview and general recommendation:
The authors investigate the effects of various seismic isolation materials between tunnels and their surrounding geoenvironment to improve protection against seismic shaking. A first comment reading carefully the article is that the paper could be of interest to the readership of the journal.
However, many points should be written more clearly, and some others should be rewritten again. I explain my points in more detail below. Thus, to make this paper publishable, the authors need to respond to the following remarks.
Major comments:
1. Lines 214-218: Authors mention a method – tool, called MIDAS/NX. Please provide a better description of this method/tool, giving also some references.
2. Line 228: Authors mention a list of soil’s material properties in Table 3. However, in Table 3 somebody can see only the properties of rock materials. Ultimately, what is the real case?
3. Line 244: It is mentioned an earthquake named El Centro happened in 1940. The readers of this manuscript may not have heard of this event. Authors should be more descriptive about this incident, by referring to where did happened, what was (briefly) the mechanism (and its magnitude), and mentioning some possible consequences on infrastructures, people, etc. In addition, which tunnel does this earthquake affect?
4. Regarding Section 4 (Results and Discussion), authors should compare the findings of their research to other researchers’ similar outcomes. They should mention some advantages and disadvantages of their research against already relevant published ones.
5. In addition, considering Section 4, and what the authors mention in lines 24-26, I would expect more analysis and further presentation regarding how geofoam, foam concrete, and silicon-based isolation material improve the protection against seismic shaking.
6. Figure 11, Figure 12, Figure 13: Make them clearer (e.g. more visible) and explain more analytically in the manuscript what they show.
7. In lines 213-218, it is mentioned that the main objective of this study has to do with soft–tunnel interactions, whereas in lines 298-301, 252-254 (as well as the title of the under-review manuscript), stresses and properties of a tunnel in rock environments are presented. Please, make clearer which geological type (e.g., soil, rock, or rockmass), the described results are associated with.
8. Section 5 (Conclusion): This section should be further discussed, meaning if the scope of this article which is mentioned in the abstract has been achieved and if the outcomes of this research can be taken into account practically in similar circumstances.
9. I would like to see more recent references about the topic of this article.
Minor comments:
10. There are many points inside the manuscript where a capital letter interferes in the middle of a sentence. For example, check that remark in the following lines: 41, 63, 98, 136, 192, 223, 269. Please correct them appropriately.
11. Line 83: What do you mean by saying rocky soil?
12. Line 117: Add the word “are” before the word “used”.
13. Line 162: Correct the word “farther” with the word “further”.
14. Line 167: What do you mean by saying “Bottom:”?
15. Line 313-314: What do you mean by saying “without liquefying by testing them with tiny shear modules or small unit weights”?
Comments on the Quality of English LanguageMinor editing of the English language is required.
Author Response
Dear Reviewer,
Thank you very much for the comments you and your eminent reviewers raised.
I have rewritten the manuscript according to the reviewers' comments. The new paragraphs and figures are highlighted in yellow.
I have also undergone English language editing by MDPI professional service.
Best regards.

Reviewer 2 Report
Comments and Suggestions for Authors
See the attached file.

The quality of the English is reasonable, but it needs to be proofread by a native speaker.
Author Response

(The authors gave the same response as above.)

Reviewer 3 Report
Comments and Suggestions for Authors
The manuscript presents a numerical study on seismic isolation materials for bored rock tunnels. In my opinion, there are many severe issues in terms of spelling, grammar, and citing format issues in the manuscript. For example, in line 45 ("Wang [24-25]"), I did not find any first authors' last names as Wang in Ref. 24 and 25. Numerous typos or spelling issues in figures (e.g., Figs. 4 and 10). The current form is not suitable for publishing in this journal.
Comments on the Quality of English LanguageExtensive English language editing is required.
Author Response

(The authors gave the same response as above.)

Round 2
Reviewer 1 Report
Comments and Suggestions for Authors
Auditing carefully the authors’ revised answers to reviewer comments, it can be clearly expressed that they have scrutinized my comments and they have integrated thoroughly their revised answers into the whole manuscript by providing (to me) satisfactory explanations.
Reviewer 2 Report
Comments and Suggestions for Authors
The authors answered the questions and corrected the paper.